# Identification of Impaired Executive Functioning after Pediatric Liver Transplantation Using Two Short and Easily Applicable Tests: Cognitive Functioning Module PedsQL and Children’s Color Trail Test

**DOI:** 10.3390/children8070571

**Published:** 2021-07-02

**Authors:** Imeke Goldschmidt, Rolf van Dick, Christoph Jacobi, Eva Doreen Pfister, Ulrich Baumann

**Affiliations:** 1Division of Pediatric Gastroenterology and Hepatology, Department of Pediatric Kidney, Liver and Metabolic Diseases, Hannover Medical School, 30625 Hannover, Germany; pfister.eva-doreen@mh-hannover.de (E.D.P.); Baumann.U@mh-hannover.de (U.B.); 2Institute of Social Psychology, Goethe University Frankfurt, 60323 Frankfurt, Germany; van.dick@psych.uni-frankfurt.de; 3Pediatric Pulmonology and Neonatology, Hannover Medical School, 30625 Hannover, Germany; jacobi.christoph@mh-hannover.de

**Keywords:** pediatric liver transplantation, cognitive functioning, cognitive impairment, school performance, CCTT, PedsQL

## Abstract

We aimed to assess executive functioning in children after liver transplantation compared with healthy controls and in relation to real-life school performance using the PedsQL^TM^ Cognitive Functioning Scale (CogPedsQL) and the Childrens’ Color Trail Test (CCTT). One hundred and fifty five children (78f, median age 10.4 (1.2–18.3) years) underwent testing with CogPedsQL and/or CCTT 4.9 (0.1–17.0) years after transplantation. Results were compared to those of 296 healthy children (165f, median age 10.0 (2.0–18.0) years). Liver transplanted children displayed significantly reduced scores for cogPedsQL and CCTT1&2 compared to healthy controls. Overall, school performance was lower in patients compared to controls. In both patients and controls, results of CCTT2 and CogPedsQL correlated strongly with school performance. In contrast to controls, school performance in patients correlated with the level of maternal but not paternal primary education degree (r = −0.21, *p* = 0.03). None of the patient CCTT or CogPedsQL test results correlated with parental school education. Conclusion: CogPedsQL and CCTT 1&2 were easily applicable in children after OLT and revealed reduced executive functioning compared to controls. Results reflect real life school performance. The association of parental education with school performance is reduced in transplanted children, which possibly indicates the overriding impact of transplant-associated morbidity on cognitive outcomes.

## 1. Introduction

Outcomes of pediatric liver transplantation have improved markedly over the past decades, with long-term survival rates around 90% [1]. With improved results quo ad vitam, the focus of long-term follow-up has now shifted to questions beyond mortality and started to address morbidity, quality of life and social inclusion after transplantation. This includes neurocognitive development and education achievements. Impairment of cognitive functioning in children after liver transplantation has been reported repeatedly [2,3,4,5,6,7,8]. Developmental delay has been described in 25–35% of children after liver transplantation [2,3]. Reduced scores for mathematics and language achievements have been reported in one third of transplanted children with normal intelligence quotient (IQ) [4]. Children after liver transplantation appear to have lower working memory abilities [7], reduced mathematic skills [5,7], lower verbal intelligence scores and worse performance in receptive language tasks [9] when compared to either a normative sample [5,7] or to children with cystic fibrosis [8,9]. Developmental delay and cognitive impairment can result from effects of the underlying primary disease [10,11,12], from transplant-associated complications [12] and from side effects of the necessary medication [5,12].

The majority of published reports on mental development, verbal performance, the general intelligence quotient (IQ) or on concentration abilities in transplanted children have used extensive psychometric testing, such as Bayley Scales, Wechsler intelligence scales or Behaviour Rating Inventories of Executive Function (BRIEF) [5,6,7,9,13,14]. While these tests offer comprehensive and differentiated understanding of a child’s impairment and cognitive functioning, they are lengthy in execution and—in the context of pediatric liver transplantation—are primarily used in research settings. Given the frequency of cognitive problems in the pediatric transplant population, a tool that offers fast and easily applicable assessment of cognitive impairment thus appears desirable. The present study presents two such instruments and provides evidence for their validity.

The PedsQL Cognitive Functioning scale (cogPedsQL) was first evaluated in children after liver transplantation in 2011 by Varni et al. [15]. This addition to the established PedsQL questionnaire on health-related quality of life consists of six questions on aspects of cognitive functioning such as selective attention, sustained attention, working memory and cognitive flexibility. It has demonstrated good discriminant validity and reliability both in children after liver transplantation and in pediatric cancer survivors [16]. The cogPedsQL comprises a patient self-report questionnaire and a corresponding parent-proxy-report. Using the cogPedsQL in liver transplanted children and their corresponding parents, cognitive functioning was found to be comparable to that of long-term cancer survivors [15], but below that of a normative population [8,15].

Second, the Children’s Color Trail Test (CCTT) is a variant of the Trail Making Test, where children are asked to connect numbers in ascending order as fast as they can. In the Children’s *Color* Trail Test CCTT, children connect numbers in ascending order in part 1, and alternate between numbers in two different colors while connecting the numbers in part 2. Using colors rather than letters for alternating attention in the second part allows cross-cultural application. Part 1 tests attention, psychomotor speed, perceptual tracking and sequencing. Part 2 also examines alternating attention, mental flexibility and response inhibition. Both the original Trail Making Test and color-trails versions such as the CCTT have detected reduced executive functioning in children with reduced academic performance, but without overtly known neurological disease [17], in children with diagnosed learning abilities, attention deficits or mild neurological conditions [18] and in HIV-positive children with a history of encephalitis [19]. One of the advantages of the CCTT is that it is comparatively quick to administer.

Given the importance of executive functioning for daily life, successful schooling and long-term quality of life, we wanted to explore the applicability of CCTT and cogPedsQL as screening instruments for reduced executive functioning in a pediatric liver transplant population. We hypothesized that the PedsQL Cognitive Functioning Module and CCTT1&2 will reveal reduced performance in children after liver transplantation compared to healthy control children. In order to assess the relevance of test results for everyday life, we also aimed to compare test performance with school performance in children of school age. External factors that might affect test results such as socio-economic and parental educational background were equally examined.

To address these questions and hypotheses, we conducted a cross-sectional study on executive functioning in liver transplanted children and in healthy control children using two tests: the Children’s Color Trail Test (CCTT), and the PedsQL Cognitive Functioning Module in a new standardized German Translation.

## 2. Materials and Methods

*Patients:* 155 children (78 girls, 77 boys) aged 1.2 to 18.3 (median 10.4) years who had undergone liver transplantation 0.1–17.0 (median 4.9) years before study entry were recruited in the outpatient clinic. No exclusions were made on the basis of primary diagnosis. Demographical data are summarized in Table 1. Parents of all children were asked to complete a proxy assessment of the Cog-PedsQL. Children aged 5 years and older completed the cognitive function module of the Pediatric Quality of Life Questionnaire PedsQL (Cog-PedsQL). Children aged 8 years and older also undertook the Children Color Trail Test (CCTT).

*Controls:* 296 healthy children (165 girls, 131 boys, aged 2.0–18.0 (Median 10.0) years) served as controls. Children below school-age were recruited from the University Hospital Day Care for staff children, which is open to all members of staff. Children of school age were recruited from a local primary and comprehensive school (“Integrierte Gesamtschule”). The number of control children was targeted at providing a European normative data set both for the new German transplantation of the Cog-PedsQL and for the CCTT.

Patients and controls had a comparable language background, with a slightly higher proportion of children with German as the primary language in the control group (88.2% vs. 81.3%, Chi square *p* = 0.064) (Table 2).

Comparison of parental education showed a similar proportion of parents without any formal degree in both groups (around 2%). There were higher rates of university entrance level diploma holders (“Abitur”) and university degree holders in the control parents compared with patients’ parents. Employment status was not documented.

*Children’s Color Trail Test (PAR Incorporated, Lutz, USA):* The Children’s Color trail test comprises 2 subtests (CCTT1 and 2). For CCTT1, children are presented with a sheet containing the numbers 1 to 15 in red and yellow circles. Children are asked to link these numbers in the correct ascending order using a pencil, working as fast as they can. The total time needed to complete the task is recorded in seconds. For the CCTT2, numbers from 2 to 15 appear twice, once in a red and once in a yellow circle. The task consists of linking the circles using the correct number sequence 1–15, while switching the color at each step, again as fast as possible. Possible errors include sequence errors in CCTT1 and both number and color sequence errors in CCTT2. Errors are pointed out by the examiner and corrected immediately, thus leading to an increase in time required to complete the task. The total time needed is recorded in seconds (raw data). In order to enable comparison of test results across age groups, PAR Inc. provides normalising calculations based on a normative population of US children. These calculations allocate a centile, a T-score and a Standard Score to any given raw time based on the age of the child. Centiles, T-score and Standard Score were calculated to describe the normative populations with a median at 50th centile, a median T-score of 50 and a median Standard Score of 100. For clarity, only Standard Scores were used for our analysis. Using the tables provided by PAR Inc., we allocated Standard Scores to the raw test results obtained by both patients and controls. CCTT tests were administered by 4 examiners who had received thorough training in test administration and followed a standardized protocol for test administration. Care was taken to ensure that CCTT was performed in quiet, undisturbed surroundings. CCTT administration took between five and seven minutes in total.

*PedsQL Cognitive Functioning Module (PedsQL^TM^, Copyright ©1998 JW Varni, Ph.D. All rights reserved):* Since the Cog-PedsQL was only available in English at the outset of our study, we undertook a standardized translation into German with the support of MAPI Research Trust, Lyon, France (https://eprovide.mapi-trust.org (accessed on 21 May 2012)). The Cog-PedsQL comprises 6 questions on the ability to concentrate in everyday life (Table A1 in Appendix A). Answers are recorded using a 5-step Likert scale. Children aged 5–7 are offered a simplified 3-step Likert scale using smiley faces. For children aged 2–4, only a parent-proxy questionnaire is used. Answers to the individual questions are scored and an average for the test is calculated. The highest possible score obtainable in the questionnaire is 100; the lowest possible score is 0. Only questionnaires where at least 4 questions had been answered were considered as valid for evaluation.

*School performance:* School performance was assessed by a questionnaire asking parents for an overall mark according to the German school marking system (1 = very good, 2 = good, 3 = satisfactory, 4 = fair/pass, 5 = poor, 6 = very poor). In addition, type of primary and secondary schooling, age at school entry and necessity to repeat grades were documented in a parental questionnaire.

*Statistical analysis:* All statistical analyses were performed using IB SPSS Statistics 25. Continuous variables are presented as mean or median as appropriate plus standard deviation. The feasibility of the cog-PedsQL was assessed by the response rate based on the number of eligible participants, by the percentage of missing values for individual items and by the frequency of complete or partial response. Crohnbach’s alpha was used to assess internal consistency reliability of the translated CogPedsQL scales. 

Results of the CogPedsQL and CCTT1&2 were compared between patients vs. controls or children vs. parents using paired or unpaired student’s *t*-test as appropriate. Effect size of significant differences between the means was determined using Cohen’s d [20]. Calculations were performed using the online calculator from https://statistikguru.de/rechner/cohens-d.html (accessed on 13 May 2021). Correlation of CogPedsQL and CCTT-results with demographic or socioeconomic factors was assessed by Pearsson’s r.

## 3. Results

### 3.1. Feasibility and Reliability of CCTT1 & 2 and CogPedsQL

Feasibility for CCTT was 100%. All children eligible for CCTT 1&2 were able to understand test instructions and completed the test.

CogPedsQL showed very good feasibility, with percentages for missing individual items of only 0.6% (children) and 0.7% (parents) respectively. 97% of participating children completed all 6 single items of the CogPedsQL questionnaire, while an additional 2.7% completed at least 5 items. Parental results were comparable, with 96.7% completion of all 6 individual items and an additional 2.8% completion of 5 items. Of 384 children eligible to complete the CogPedsQL, 15 refused participation. Fourteen of these were in the 5–7 age group, highlighting the difficulties of recruiting younger children. 

Parental response rate was 95% and was mainly influenced by parental availability during hospital or school visits. Crohnbach’s alpha as a measure for internal consistency reliability was 0.74 for the children’s CogPedsQL and exceeded 0.9 for the parental CogPedsQL. 

### 3.2. Children’s Color Trail Test in Patients and Controls

Results for CCTT1&2 are summarized in Table 3. In patients, mean Standard Scores were 81.5 (54–120) for CCTT1 and 87.8 (54–115) for CCTT2. Analysis according to age group (8–12 vs. 13–16 years) revealed significantly better results in the older patients for CCTT1, but not for CCTT2. This difference between age groups was not present in the control children.

Transplanted children scored significantly lower than healthy controls both in CCTT 1 and 2, with an average effect size of d = −0.84 (Table 3). Test results in patients appear to follow a bimodal distribution, with CCTT2 results that indicate “no impairment” of executive functioning displaying a near normal distribution. However, there appears to be a second peak in the “impairment” range (Figure 1a), in contrast to the control results. Impairment is predominantly mild (Figure 1b).

### 3.3. CogPeds-QL

Valid test results of CogPedsQL were available from 121 patients and 147 parents of transplanted children. Mean patient score was 66.8 (range 8.3–100), while mean parent-proxy report score was 65.2 (range 4.2–100). Paired comparison of children’s and parents’ results was possible in 116 patients. Overall, patients trended to judge their executive functioning better than their respective parents (mean CogPedsQL score 67.3 vs. 63.8, *p* = 0.11), although this difference became significant only in the 8–12-year-olds (59.2 vs. 55.9, *p* < 0.01).

Patients scored significantly lower than controls with medium effect size in the children’s reports (d = −0.39), but strong effect size in the parent-proxy reports (d = −0.73). However, this difference could not be found in all age groups (see Table 4). Parent-proxy scores were almost identical between patients and controls in the toddler/preschool group (2–4 years old). In contrast, children’s scores did not differ significantly in the teenage age group (13–18 years).

### 3.4. Influence of Age and Socio-Economic Factors on Test Results

CogPedsQL self report scores for 5–7 year olds were significantly lower than those of all other age groups (50.5 in patients, 61.6 in control children (*p* = 0.046, d = −0.49)). This difference is not reflected in the corresponding parent-proxy reports (Table 4). Corresponding to the lower CogPedsQL results in the 5–7 age group, age at testing shows a significant correlation with children’s CogPedsQL (r = 0.27, *p* < 0.01), which disappears if the results for 5–7 year olds are eliminated from the analysis (r = 0.02, *p* = 0.91).

CCTT1 showed a weak, but positive correlation with age at testing (r = 0.13, *p* = 0.028), but no correlation of age with CCTT2 or parental CogPedsQL could be demonstrated.

Socio-economic background appears to be associated with test performance in control children. CogPedsQL scores show weak, but consistent and statistically significant correlation with parental education in control children (Table 5). This association is much less pronounced in transplanted children, where only maternal tertiary education status is associated with CogPedsQL results. 

### 3.5. Results of Children’s Color Trail Test and CogPedsQL Are Reflected in School Performance

A hundred and one patients were attending school at the time of our study. Thirty-one were in primary school, and 7 attended a special needs school. More patients than controls had entered school belatedly at age 7 years rather than at the usual 6 years (patients vs. controls 6 years 62% vs. 78.5%; 7 years 33.3 vs. 13.3%, *p* < 0.01 respectively). Twenty-eight patients (26.4%) had had to repeat a grade, the majority of which (72%) were in primary school. 

Of 63 patients in secondary schooling (10 years and above), 7 (11.1%) attended basic level secondary schools, 29 (46.0%) attended mid-level schools and 15 (23.8%) attended higher level secondary education (Gymnasium) leading to a university entrance diploma. Twelve (19.0%) attended secondary schools offering several degree levels. These proportions are markedly shifted compared to the general population in Germany, with a higher proportion attending mid-level schooling and a lower proportion achieving university entrance level in the patients (Figure 2a).

School performance was rated as overall mark by the parents. There was a marked difference in school performance between patients and controls (Figure 2b). Only 4.7 % of patients were rated to achieve “very good” results, compared to 23.2% of controls. Rates for good/satisfactory/fair/poor and very poor were 35.9%/41.5%/14.2%/2.8%/0% in the patients and 56%/17%/3.7%/0%/0% in the controls (*p* < 0.001).

In both patients and control children, results of CCTT2 and cogPedsQL correlated significantly with school performance (CCTT2 r = −0.29/−0.19 in patients/controls respectively, children’s CogPedsQL r = −0.32/−0.27, parental CogPedsQL r = −0.32/−0.57 respectively, negative values for correlations as school performance is inversely marked).

At the same time, school performance of control children showed a medium effect size correlation with their parents’ education status (Table 6). In patients, only maternal secondary education status correlated with school performance (r = −0.21, *p* = 0.03). 

## 4. Discussion

We examined cognitive functioning in children after liver transplantation by using two simple and short tests that can be used in everyday clinical practice. The Cognitive Functioning Module of the PedsQL (cogPedsQL) tests the ability to concentrate and maintain concentration (cognitive fatigue), while the CCTT examines executive functioning such as attention, psychomotor speed, perceptual tracking, sequencing, alternating attention, mental flexibility and response inhibition.

We found reduced scores for both the cogPedsQL and CCTT in liver transplanted children when compared to a healthy control sample.

The Cognitive Functioning Module of the PedsQL was first published in the context of pediatric liver transplantation by Varni et al. in 2011 [15], who found reduced CogPedsQL scores in 215 liver transplanted children and their parents compared to a corresponding normative sample [15]. Varni et al. also obtained the 72-item Behaviour Rating Inventory of Executive Function (BRIEF) on their liver transplant cohort. They could show strong and significant correlations of cogPedsQL results with all BRIEF subscales, supporting the validity of the cogPedsQL as a measure for cognitive fatigue and executive functioning. More recently, Ohnemus et al. used the cogPedsQL in a longitudinal assessment of health-related quality of life (HRQOL) in children and adolescents after liver transplantation [8]. They also documented reduced cognitive functioning as measured by the cogPedsQL when comparing liver transplanted children with a healthy normative sample.

The validated German translation of the cogPedsQL was created by our group in cooperation with MAPI research trust specifically for this study. It has since been used by Petersen et al. as an integral part of the PedsQL fatigue scale, with similar results for the transplanted children [22]. Results for cogPedsQL in our study appear lower than in Varni’s and Ohnemus’ publication using the English original. Similarly, healthy control cogPedsQL results in our study are lower than, for instance, those in the normative US sample used in the study by Petersen et al. [22]. While the new German translation followed a standardized, evaluated process to ensure comparability with the English original, it is conceivable that the translation per se causes a systematic shift in answers. Also, cross-cultural differences that are independent of language might have an impact on the way individual questions are understood and answered. Differences in results according to country of origin despite the use of the same language have previously been described for the generic PedsQL [23]. This observation underlines the importance of matching, language-identic control cohorts. Our study presents the first German-speaking pediatric control cohort for the cognitive functioning module of the PedsQL. 

The CCTT as such has not yet been applied in children after liver transplantation. However, a trail making test was used in long-term survivors after pLTx as part of the Delis-Kaplan Executive Function System (D-KEFS) battery [24]. Here, reduced scores were found in the transplanted children compared with their healthy siblings. We chose the CCTT because it is fairly quick to administer. Also, CCTT appears less influenced by primary language than the Trail Making Test [25]. Its results appear to be independent of gender [25], but need to be normalized for age. 

For CCTT, the importance of an adequate local normative sample for comparison has also been demonstrated [26], as there appear to be more differences based on sociocultural background than previously anticipated. Our normative sample was comparable with the group of transplanted children in its composition and diversity with regards to language and migration background. Our data represent the first study of CCTT in liver transplanted children, as well as delivering normative data for a European population.

In our study, results of both tests correlated with school performance, thus supporting the relevance of test results for everyday life. We found school performance in general to be lower in transplanted children compared with healthy controls. Some inaccuracy must be assumed, since instead of objective itemized scoring of academic performance, we used global parental assessment as a rather crude measure for school performance. However, the fact that significantly more of our patients receive basic level and mid-level secondary education than is currently seen in the general population in Germany supports the results of parental assessment. The rate of children attending a special needs school in our cohort (7%) is lower than published rates of 10–34% [4,7,27]; however, different school systems make it difficult to compare results directly.

Some caveats need to be applied when using the tests in daily clinical practice:

For cogPedsQL, we saw significant differences between the different age groups in the patient cohort. In 5–7-year-olds, the 5-point Likert scale for answering the cogPedsQL questions is replaced by a 3-point smiley face scale (smiley/neutral/grumpy face). The smiley face signifies to “never” have any problems. It is conceivable that a negative bias is introduced by the reduction to 3 options of answering a question. This interpretation is supported by the fact that reduced scores in the 5–7-year-olds were also observed in the control group. Based on these results, we would caution against using the cogPedsQL in the 5–7 years age-group in order to avoid overestimation of cognitive fatigue. 

A similar caution applies to the use of the cogPedsQL in the 2–4-year-olds. Parents frequently reported difficulties in filling in the questionnaire for this age group, as some of the items appeared difficult to assess in 2–4-year-olds. This might lead to an over-positive assessment of cognitive abilities in this age-group. Some of the difficulties in concentration observed in this age group might still be judged as age-appropriate by the parents, potentially leading to over-estimation. This notion is supported by the fact that parent-proxy reports of patients scored highest in the 2–4-year-olds and in fact were not significantly different from control values.

CogPedsQL results in controls were correlated with parental education status, which we used as surrogate parameter for socioeconomic background. This finding highlights the strongest source of potential bias in our study, namely the somewhat uneven distribution of socioeconomic levels between controls and transplant cohort. We tried to minimize that source of bias by our choice of school and day care centre for control recruitment; however, there still is a higher proportion of university entrance level diploma holders and university degree holders among parents of the control group. One explanation might be that consent for study participation among controls was influenced by higher socioeconomic status. The perceived association of test results and parental education background raises the question of validity for the difference in test performance between patients and controls. Multivariate regression identified both maternal education level and patient vs. control status as independent predictors of PedsQL results, albeit with considerably higher beta coefficient for health status. We therefore believe that PedsQL can be used as a valid instrument to compare transplanted and healthy children, if interpretation is made in light of variable socioeconomic backgrounds.

Similar to test performance, school performance of control children correlated significantly with their parents’ education status. This effect was much less pronounced in patients. Correlation of academic achievement and socio-economic status has been repeatedly reported for Germany as well as for other European countries (20). Why these effects were not observed in the transplant cohort remains an open question. Published results on a link between socioeconomic status and cognitive abilities in children with liver disease or liver transplants are equivocal. Most studies comparing patients and controls matched controls according to age, gender and ethnicity [15,28,29], but not to socioeconomic background. One study described higher household educational levels to be a protective factor for mental development at 2 years of age [27], while others could not delineate differences in cognitive abilities in liver transplanted children according to socioeconomic status [24,30,31]. We have previously shown that age at transplantation and length of stay in intensive care are the main predictors of cognitive outcome in children after liver transplantation [32]. We therefore interpret the observed lack of association with socio-economic background in this analysis as a result from overriding influences of somatic factors.

## 5. Conclusions

In summary, we describe the cognitive functioning module of the PedsQL and the CCTT as two easily applicable tests that can detect impaired executive functioning in children after liver transplantation. Test results correlate with school performance, indicating their relevance for daily life. The influence of parental education on school performance is reduced in transplanted children, which possibly indicates the overriding impact of transplant-associated morbidity on educational outcomes. We deliver the first application of the German translation of the cogPedsQL, including data for a normative control group. We also deliver control data for the CCTT from a European cohort. Our data support the fact that cogPedsQL and CCTT can be used in clinical practice to identify children who might benefit from more detailed neuropsychological assessment and academic support. 

## Figures and Tables

**Figure 1 children-08-00571-f001:**
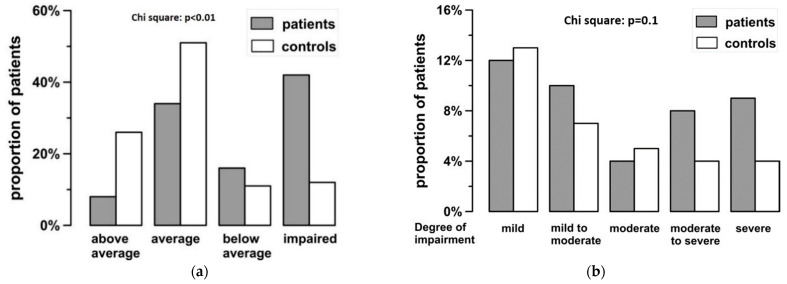
(**a**) Results of CCTT2 are classified into categories according to their Standard Score [21]. Standard Score ≥ 107 equals above average performance, while Standard Scores of 92–106 and 85–91 are classed as average and below average respectively. Standard Scores below 85 are classed as “impaired”. Shaded bars depict patients, white bars depict controls. (**b**) “Impairment” can be further differentiated by Standard Score: 77–84 mildly impaired, 70–76 mild-to-moderately impaired, 62–69 moderately impaired, 55–61 moderately-to-severely impaired, 0–54 severely impaired. Shaded bars depict patients, white bars depict controls.

**Figure 2 children-08-00571-f002:**
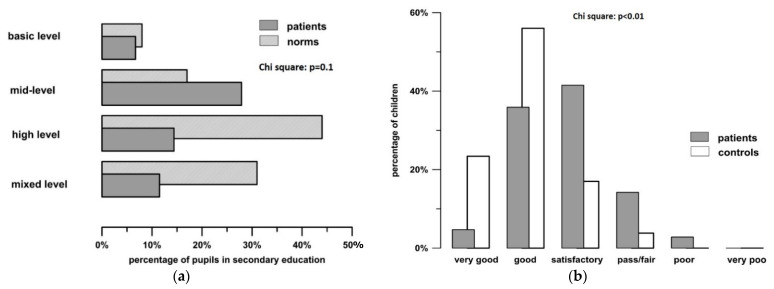
(**a**) Comparison of type of secondary schooling between transplanted patients and norms. Norm data in this figure are drawn from official German census data from 2016 including all children visiting secondary schooling in Germany. Patients are depicted by grey bars, and norm data is depicted by hatched bars. (**b**) School performance of patients (shaded bars) compared to controls (white bars) as rated by their parents.

**Table 1 children-08-00571-t001:** Demographic data of the study population.

			*n* (%)/Median (Range)
**Sex**	Patients	Boys	77 (49.7%)
		Girls	78 (50.3%)
	Controls	Boys	131 (44.3%)
		Girls	165 (55.7%)
**Age**	Patients		10.4 years (2.1–18.3)
	Controls		10.0 years (2.0–18.0) n.s.
**Primary disease**			
BA			81 (52.3%)
Acute liver failure			11 (7.1%)
Alpha-1-antitrypsin-deficiency			9 (5.8%)
M. Wilson			3 (1.9%)
AIH/PSC			7 (4.5%)
PFIC			7 (4.5%)
Alagille-Syndrome			8 (5.2%)
Hepatoblastoma			5 (3.2%)
Metabolic disease			
Other (including: ARPKD (2), CF (6), neonatal hemochromatosis (2), Budd–Chiari syndrome (1), neonatal cholestasis of unknown reason (2), portal vein thrombosis (1), IFALD (1), glycogenosis (2), OTC deficiency (1), primary hyperoxaluria (1), choledochal cyst (1), and cryptogenic cirrhosis (4))			24 (15.5%)

BA: extrahepatic biliary atresia, AIH: autoimmune hepatitis, PSC: primary sclerosing cholangitis, PFIC: progressive familial intrahepatic cholestasis, ARPKD: autosomal recessive polycystic kidney disease, IFALD: intestinal failure associated liver disease, CF: cystic fibrosis, OTC: ornithine-transcarbamylase.

**Table 2 children-08-00571-t002:** Socioeconomic and cultural background of study population.

	Patients *n* (%)	Controls *n* (%)	*p* (Chi Sqare)
**Country of birth**			
Germany	150 (96.8%)	284 (95.9%)	n.s.
Outside Germany ^1^	1 (0.6%)	5 (1.6%)	
Information missing	4 (2.6%)	7 (2.4%)	
**Native Language**			
German	126 (81.3%)	261 (88.2%)	n.s.
Turkish	4 (2.6 %)	11 (3.7%)	
Russian	4 (2.6%)	3 (1.0%)	
Other ^2^	13 (8.3%)	13 (4.4%)	
Information missing	8 (5.2%)	8 (2.7%)	
**School Leaving Certificates**			
Mothers			
None	2 (1.3%)	4 (1.4%)	
Basic level	21 (13.5%)	17 (5.7%)	<0.01
Mid-level	60 (38.7%)	81 (27.4%)	
University entrance level	53 (34.2%)	180 (60.8%)	<0.01
Other	10 (6.4%)	5 (1.7%)	
Information missing	1 (0.6%)	9 (3.0%)	
Fathers			
None	3 (1.9%)	5 (1.7%)	
Basic level	32 (20.6%)	16 (5.4%)	<0.01
Mid-level	41 (26.5%)	61 (20.6%)	
University entrance level	57 (36.8%)	180 (60.8%)	<0.01
Other	8 (5.2%)	5 (1.7%)	
Information missing	1 (0.6%)	29 (9.8.0%)	
**Highest Professional Degree in the Family**			
None	7 (4.5%)	10 (3.4%)	
Apprenticeship	40 (25.8%)	47 (15.9%)	<0.01
Vocational school	39 (25.1%)	46 (15.6%)	<0.01
University of cooperative	17 (11.0%)	41 (13.9%)	
**Education**			
University degree	24 (15.5%)	81 (27.4%)	<0.01
PhD	10 (6.5%)	55 (18.6%)	<0.01
Other	6 (3.9%)	1 (0.3%)	
Information missing	12 (7.7%)	15 (5.1%)	

^1^ other country of birth includes Azerbaijan for 1 patient and USA (*n* = 3), Israel and Uruguay *n* = 1 each for controls. ^2^ Other native languages include: patients: Portuguese and Urdu for *n* = 2 respectively and Kurdish, Albanian, Afghan, Vietnamese, Sinti, Moroccan, Polish, Azerbaijanian and Spanish for *n* = 1 resp.; controls: Kurdish (*n* = 6), Macedonian and Albanian (*n* = 2 resp.) as well as Cantonese, Serbian and Spanish (*n* = 1 resp.).

**Table 3 children-08-00571-t003:** Results of children’s color trail test (CCTT) in transplanted children and healthy controls.

	Patients	Controls	*p*	Cohen’s d
**CCTT1**				
All	81.5 ± 19.3 *n* = 84	95.3 ± 15.1 *n* = 191	<0.01	−0.84
age 8–12	77.1 ± 17.8 *n* = 40	94.2 ± 15.1 *n* = 115	<0.01	−1.1
age 13–16	85.5 ± 19.9 *n* = 44	96.9 ± 15.2 *n* = 76	<0.01	−0.67
	8–12 vs. 13–16 *p* = 0.04 d = −0.44	8–12 vs. 13–16 *p* = 0.22 n.s.		
**CCTT2**				
All	87.8 ± 15.6 *n* = 85	98.7 ± 11.9 *n* = 191	<0.01	−0.83
age 8–12	86.9 ± 16.3 *n* = 40	99.3 ± 11.3 *n* = 115	<0.01	−0.97
age 13–16	88.7 ± 15.1 *n* = 45	97.1 ± 12.7 *n* = 76	<0.01	−0.62
	8–12 vs. 13–16 *p* = 0.6 n.s.	8–12 vs. 13–16 *p* = 0.14 n.s.		

**Table 4 children-08-00571-t004:** Results of PedsQL cognitive functioning module in transplanted children and healthy controls.

	**Patients** **cog-PedsQL Children**	**Controls** **cog-PedsQL Children**	***p*** **Patients vs. Controls**	**Cohen’s d**
**PedsQL Children**				
All	66.8 ± 20.5 (*n* = 121)	73.9 ± 17.1 (*n* = 247)	<0.01	−0.39
age 5–7	50.5 ± 22.0 (*n* = 24)	61.6 ± 22.6 (*n* = 54)	0.046	−0.49
age 8–12	68.9 ± 19.5 (*n* = 42)	78.1 ± 13.5 (*n* = 115)	<0.01	−0.60
age 13–18	72.4 ± 16.8 (*n* = 55)	76.4 ± 13.3 (*n* = 78)	0.13 n.s.	−0.27
5–7 vs. 8–12	*p* < 0.01, d = −0.90	*p* < 0.01, d = −0.98		
5–7 vs. 13–18	*p* < 0.01, d = −1.18	*p* < 0.01, d = −0.84		
8–12 vs. 13–18	0.34 n.s., d = −0.19	0.39 n.s., d = 0.13		
	**Patients** **Parent proxy cogPesQL**	**Controls** **Parent proxy cog-PedsQL**	***p*** **Patients vs. Controls**	
**PedsQL Parents**				
All	65.2 ± 23.5 (*n* = 147)	79.2 ± 16.6 (*n* = 279)	<0.01	−0.73
Age 2–4	74.7 ± 22.5 (*n* = 25)	75.5 ± 14.2 (*n* = 29) *	0.85 n.s.	−0.04
age 5–7	61.9 ± 21.1 (*n* = 29)	80.2 ± 15.7 (*n* = 55) *	<0.01	−1.03
age 8–12	55.7 ± 22.8 (*n* = 39)	80.7 ± 16.1 (*n* = 110) *	<0.01	−1.38
age 13–18	69.3 ± 23.3 (*n* = 54)	79.17 ± 18.2 (*n* = 74) *	0.01	−0.48
2–4 vs. 5–7	*p* = 0.035, d = 0.59	*p* = 0.18 n.s.		
2–4 vs. 8–12	*p* = 0.002, d = 0.84	*p* = 0.11 n.s.		
2–4 vs. 13–18	*p* = 0.34 n.s.	*p* = 0.33 n.s.		
5–7 vs. 8–12	*p* = 0.26 n.s.	*p* = 0.83 n.s.		
5–7 vs. 13–18	*p* = 0.15 n.s.	*p* = 0.74 n.s.		
8–12 vs. 13–18	*p* = 0.006, d = −0.59	*p* = 0.55 n.s.		
	**Patients** **Paired cog-PedsQL** **Children vs. Parents**	**Controls** **Paired cog-PedsQL** **Children vs. Parents**		
All	67.3 ± 20.3 vs. 63.8 ± 22.7 (*n* = 116) *p* = 0.11, n.s.	74.3 ± 16.9 vs. 79.9 ± 16.7 (*n* = 231) *p* < 0.01, d = −0.33		
age 5–7	51.2 ± 22.2 vs. 65.9 ± 16.1 (*n* = 23) *p* = 0.012, d = 0.76	61.7 ± 22.4 vs. 79.2 ± 16.0 (*n* = 50) *p* < 0.01, d = −0.89		
age 8–12	69.2 ± 19.9 vs. 55.9 ± 23.1 (*n* = 38) *p* < 0.01, d = 0.62	78.6 ± 13.0 vs. 80.5 ± 16.1 (*n* = 109) *p* = 0.26 n.s.		
age 13–18	72.9 ± 16.4 vs. 69.3 ± 23.5 (*n* = 53) *p* = 0.16, n.s.	76.4 ± 13.4 vs. 79.3 ± 18.1 (*n* = 72) *p* = 0.19, n.s.		

* In 11 control children, age was not documented.

**Table 5 children-08-00571-t005:** Correlation of executive functioning and parental formal education background–Pearson’s rho correlation coefficient.

	Mother		Father	
	Level of Secondary Education	Level of Tertiary Education	Level of Secondary Education	Level of Tertiary Education
**CCTT1**				
Patients	0.1	0.05	0.19	0.25
Controls	0.02	0.00	0.00	−0.05
**CCTT2**				
Patients	0.09	−0.02	0.06	0.05
Controls	0.04	0.11	0.12	0.17
**cogPedsQL Children**				
Patients	0.09	0.27 **	0.11	0.12
Controls	0.16 **	0.14 *	0.14 *	0.16 *
**cogPedsQL Parents**				
Patients	0.05	0.18 *	0.05	0.07
Controls	0.26 **	0.19 **	0.24 **	0.22 **

* *p* < 0.05 ** *p* ≤ 0.01.

**Table 6 children-08-00571-t006:** Correlation of school performance and parental formal education background–Pearson’s rho correlation coefficient. School marks range from 1: very good to 6: very poor, i.e., lower numbers signify better performance.

	Mother		Father	
	Level of Secondary Education	Level of Tertiary Education	Level of Secondary Education	Level of Tertiary Education
School performance				
Patients	−0.21 *	−0.03	−0.11	−0.05
Controls	−0.26 **	−0.26 **	−0.26 **	−0.21 **

* *p* < 0.05 ** *p* ≤ 0.01.

## Data Availability

Data are available from the corresponding author on request.

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
