# Peer review of "Identification of Impaired Executive Functioning after Pediatric Liver Transplantation Using Two Short and Easily Applicable Tests: Cognitive Functioning Module PedsQL and Children’s Color Trail Test"

_children, 2021, doi:10.3390/children8070571_

Round 1
Reviewer 1 Report
The paper is well written and presents two tools to evaluate the cognitive functioning after LT in the pediatric population.
I would suggest:
- In the figure, to add the p-value, to better understand the difference among the groups.
- it would be of interest to evaluate the tests in the population according to the etiology and the indication to LT. As the paper describe, some diseases have effects on cognitive function.
Author Response
Thank you for the valuable comments!
We have added p-values in figures 1a, 1b and 2a, 2b. Please note that for these figures, calculations of difference between patients and controls was made by chi square test for the distribution of categorical variables. Therefore, there is one p value per figure, not several p values for individual column comparison.
You also suggested that we look at potential differences in cognitive functioning between patients according to their primary disease. While we agree that this is an interesting proposition, we feel that our dataset is not suitable to make these analyses. 52% of our patients had biliary atresia. The remaining 48% of patients are spread across 20 different disease entitities. We fear that the resulting group sizes preclude any meaningful analysis. A larger, preferably multicentric patients sample might answer this question. We have shown in a previous manuscript (Goldschmidt et al, JPGN 2019 Apr;68(4):480-487, PMID 30664562) that low age at transplantation and prolonged stay in the ICU lead to decreased cognitive functioning, and of course there is a large proportion of BA patients in the low age at transplant group. We hope to follow up on this topic in a future multicentric project.
Reviewer 2 Report
This is a very well performed study with high clinical relevance for pediatric patients after liver transplantation. The manuscript is very well written with no obvious flaws. The results are presented logically and data interpretation is appropriate.
Author Response
Thank you for the kind review